# Configuration-Based Promotion: A New Approach to Destination Image Sustainability

**Yanan Li** [1], **Honggen Xiao** [2], **Naipeng Bu** [3,*], **Jianji Luo** [2], **Hui Xia** [3], **Liyuan Kong** [3] and **Haoyue Yu** [3]

1   School of Innovation and Management, Suan Sunandha Rajabhat University, Bangkok 10300, Thailand; handanlyn@sina.com
2   School of Hotel and Tourism Management, The Hong Kong Polytechnic University, Hong Kong, China; honggen.xiao@polyu.edu.hk (H.X.); jianji.l.luo@connect.polyu.hk (J.L.)
3   Business School, Shandong University, Weihai 264209, China; Grace.xia@sdu.edu.cn (H.X.); zyl520kly@163.com (L.K.); yuhaoyue325@163.com (H.Y.)
*   Correspondence: bunp@sdu.edu.cn

**Abstract:** The assessment of tourist destination images should not only be the arrangement of multiple influencing factors. This study explores the complex causal relationship for tourist destination images based on a configuration perspective to enhance the overall tourism image using the fuzzy-set qualitative comparative analysis method. The configurational paths for promoting tourism image can be categorized into two types and nine configurations in Shandong Province. Therefore, this study recommends augmenting the tourism image of the entire region with the logical thinking of "segmentation–integration" and "enhancing strengths–supplementing weakness" and finally realizing a sustainable tourist destination image.

**Keywords:** sustainable tourist destination image; fsQCA method; configuration path; Shandong Province

## 1. Introduction

The two characteristics of tourist destinations are that they are intangible products and that they are difficult to try before buying, which render the destination image the primary competitive tool available between destinations [1]. Destination image exerts a crucial effect on tourists' decision making, which has been empirically established [2]. Thus, many tourist destinations consider creating a positive and attractive image as the core objective of their marketing [3]. Positive image building is a key element to attract tourists, stimulate their curiosity and sense of pursuit, and ultimately compels them to make travel plans. In addition, it plays a key role in travel decision making, in-travel experience, and post-travel evaluation. Furthermore, it is the soul and core of tourism development. As the business card and intangible asset of tourist destinations, it is one of the main driving forces for the promotion of tourism development [4].

The vigorous growth of the tourism market will inevitably create intensified competition within it. Under such fierce competition, "image" plays a critical role in competition and collaboration, directly affecting its comprehensive sustainable development. If a tourist destination intends to have an absolute competitive advantage, it must have an excellent destination image in the face of fierce industry competition. Given that the research on tourist destination image was first proposed by American scholar Gunn in the 1970s, it has been a crucial field of tourism research by scholars [5]. In China, for example, tourism has been studied since the 1990s. Finally, the research content involved several aspects such as the concept of the destination image, formation mechanism, life cycle, measurement, and evaluation [6], which are still a hotspot in tourism research. In the previous literature on tourist destination image, few studies have examined it from the standpoint of configuration, and the segmentation of a complete tourist destination with the logical thinking of "segmentation–integration" and "enhancing strengths–supplementing weakness" has

yet to be explored. This study will fill the gaps in the research in these aspects. This study answers the applicability of the fuzzy-set qualitative comparative analysis (fsQCA) method in the field of sustainable tourist destination image, which not only broadens the applicable research scope of this method but also demonstrates that a sustainable tourist destination image can be summarized into equivalent promotion paths to uphold the sustainable development of the whole tourist destination.

If a province is considered a complete tourist destination, then each city in this province can have its own tourism development strategy. Thus, the motivation of this study is to take a province as an overall research object to break the state of each city running its own affairs. The purpose is to determine the advantages and disadvantages of each city in a province to give full play to the advantages and evade the disadvantages of each group of cities. Furthermore, the objective is to finally realize the improvement of the overall tourism image of a province and the sustainable development of tourism.

## 2. Literature Review

The content of the research on tourist destination image is extensive, including concepts, formation process, influencing factors, dimensions, evaluation methods, and sustainable development; the latter three are more suitable for the present study. Thus, this work reviews the literature from these aspects.

### 2.1. Dimensions of Tourist Destination Image

The image dimension of a tourist destination was first categorized as the native image and induced image by Gunn [5]. The basis for judging the abovementioned two images was whether tourism resources were disturbed by business information and whether the cognition of tourist destinations has been processed. Huang and Li [7] categorized the dimensions into original images and induced images, which are the same as Gunn's dimensions but with different meanings. On the basis of Gunn's research, Fakeye and Crompton [8] proposed a complex image. A compound image increases the factor of personal experience, particularly the impression of the destination obtained by tourists through personal experience. Later, other dimensions have been proposed, such as the three-dimensional characteristics of the pairwise relative, namely, "function–psychology", "element–integration", and "commonality–individuality" [9]; launching destination image and receptive destination image [10]; cognitive, affective, and intentional images [11]; image formation, market segmentation, and competitive strategy [12]; stereotyped, affective, and unique images [13]; background and strengthened images [14]; cognitive, affective, and the overall images [15]; cognitive image, affective image, and marketing communication, which are considered from the perspective of travel agency marketing [16]; and projected and perceptual images, which are considered from the reception effect of the source market [17]. With the emphasis on air quality, the terrain image of tourist destination is categorized into three dimensions of perception of smog: perceived health, traffic and tourist experience risk, and revisit intention [18].

### 2.2. Evaluation Method of Tourist Destination Image

Other scholars have divided the image measurement methods of tourist destinations into structural and nonstructural methods [7]. Structural methods use different evaluation factors, whereas nonstructural methods use free questionnaires to record the interviewees' descriptions of the destination image. The evaluation methods for tourist destination images can also be categorized into qualitative and quantitative methods. Qualitative methods are as follows: free heuristics method, which uses text connotations to obtain subjective knowledge of products [19]; content analysis method, which uses graphic information to examine destination images [20]; and multimodal discourse method, which uses construction to research the tourist image [21]. Conversely, the quantitative methods are as follows: cognitive mapping methods, which can better reflect the direction of the destination in the human brain, the surrounding environment, and the spatial correlation [22];

correspondence analysis method, which describes the correlation between two or more categorical variables at each level through principal component analysis [23]; intermediary test method, which verifies the antecedents in tourism image [24]; and mathematical models, such as regression analysis, importance–performance analysis (IPA), tri-component model [25–27], structural equation modeling (SEM) [28], and SEM in tourism [29,30]. Other studies have combined relevant theories for research, such as tourism discourse theory, grounded theory, and cultural memory theory [31–33]. Other studies have used the GIS platform to analyze the attraction of tourist destinations from the viewpoint of spatial characteristics [34]. With the progress of information technology, research has been conducted on smart destination image. The evaluation methods include SEM, network big data, and deep learning algorithms [35–37].

### 2.3. Sustainable Tourist Destination Image

For sustainable tourist destination image, the research angles are relatively diverse. The ecological "Ceylon tea" brand should be used to create a destination image, and sustainable tourism experience packages should be launched [38]. A right market positioning plays an important role in the sustainable tourism development in Russia [39]. The network relationship can better integrate and harness destinations' resources to promote sustainable tourism development in urban agglomerated areas [40]. Data analysis methods, including exploratory factor analysis, confirmatory factor analysis, and SEM, have been used in rural tourist destination research [41]. A destination loyalty model is used to predict tourists' willingness to choose a sustainable tourist destination [42]. Statistical logit and probit models are used to call on tourist destinations to promote sustainable tourism development through investments and initiatives [42]. City branding focuses on creating an image and on the desire to create and form positive associations with the city [43].

### 2.4. Research Review

Many research directions are present in the dimensions of the tourist destination image. Per different research angles, the classification of dimensions also differs. Two research directions are as follows. One is the view of Gunn, the earliest proponent of the topographic image dimension of tourist destinations; his view has been recognized by most scholars, including Fakeye, Crompton, Huang, and Li [7,8]. Second is identifying another angle to develop a new research direction from and no longer continues Gunn's classification standard. Studies on the dimensions of tourist destination image have proposed various classification methods, all of which have specific applicability. From the standpoint of discipline integration, scholars have proposed more research angles and showed the trend of multidisciplinary integration, such as literature, aesthetics, psychology, neural network, economics, mathematics, geography, computer science, and other disciplines. From the attribute of the research method, the research methods no longer satisfy the qualitative research represented by text analysis but align more with the combination of qualitative and quantitative methods. The representative method is the combination of network text analysis and the IPA model. For sustainable tourist destination image, although the research angles are relatively diverse, such as ecological tea brand, right market positioning, tourists' willingness, and urban agglomeration system, the results of the study in quantitative or qualitative analysis are relatively singular.

Despite fruitful and mature research on tourist destination image, some limitations persist in the existing research. This study aims to breakthrough these limitations, focus on the elements generally overlooked in destination image research, and incorporate them into the research design. First, the research content is relatively single, boundaries are relatively clear, and documents with mixed research on concepts, dimensions, and evaluation methods are few. Second, the research angle is primarily static effect analysis, rather than variable combination analysis.

## 3. Research Methods

The direct motivation of qualitative comparative analysis (QCA) lies in people's curiosity about solving the phenomenon of causal complexity; sociologist Ragin pioneered this direction [44]. The QCA method is between variable and case-oriented approaches, surpassing the new method of qualitative and quantitative research; it is a method of classifying cases based on theoretical concepts, and the research problem reflects the configuration problem rather than the net effect problem. In the practice path, this method reflects the collective relationship rather than the correlation relationship [45]. fsQCA is a type of QCA method. As the original data generally cannot fulfill the analysis requirements, it must be transformed into an explicit set or a fuzzy set when assigning values to all variables. The fundamental principle of this method is the overall analysis of the case set [46], which can address the complex interactive correlation between multiple influencing factors, as well as analyze the combination of multiple antecedent variables of specific results [47]. On the basis of this method, multiple equivalent configuration schemes that produce the same result can be identified, and more path options for attaining success or avoiding failure based on its conditions can be provided. The application of the QCA method will make revolutionary contributions to the tourist destination image in solving complex causal relationships and improving practical relevance.

## 4. Research Process

### 4.1. Case Selection

The GDP of Shandong Province ranks third in China, its population ranks second, and its overall scale of tourism development ranks fourth. Shandong Province is the birthplace of ancient Chinese culture (the Dawenkou and Longshan cultures) and is the hometown of Confucius. The province has vibrant tourism resources, including four world heritages and eight world intangible cultural heritages. In addition, 17 cities in Shandong Province have their own characteristic tourism resources, such as Confucius Culture in Ji'ning City, Spring Water in Ji'nan City, Mount Tai in Tai'an City, Qi Great Wall in Laiwu City, and so on. Accordingly, Shandong Province is a suitable research object for the configuration analysis. Hence, we selected all 17 cities in Shandong Province as the research area. The tourism data differed from previous years due to the influence of COVID-19 in China at the end of 2019; thus, we selected 2018. Notably, Laiwu City was included in the provincial capital Jinan as a district in October 2018; however, Laiwu area is still considered a city in the annual statistical yearbook. To reflect the full "comparison" between cases, case selection should pursue the principle of maximum heterogeneity—including positive and negative cases [48]. Hence, Laiwu area was considered a separate case for analysis in this study.

### 4.2. Variable Calibration

We established a direct link between the image dimension of tourist destinations and the antecedent variables (dimensions were antecedent variables). Starting from the latest research results, the dimensions included three basic dimensions, namely, cognitive, affective, and composite images [49]. Per the variable setting of the QCA method, it was categorized into outcome and antecedent variables. In this study, total tourism revenue was used as an indicator to measure the tourist destination image, that is, the total tourism revenue was set as the result variable. In addition, we set the dimensions of the tourist destination image as antecedent variables, which corresponded to the native, induced, and composite images and then refined the three antecedent variables, that is, the construction of secondary indicators. Based on the classification of tourist destination image, the secondary index of the native variable was refined into geographical and cultural variables [50]. The design of the induced variable is based on the six elements of tourism and the principle of data availability from the official website [45]. Therefore, the induced variable selects the three tourism elements of "diet, accommodation, and traffic". The element "diet" is explained by the weighted score of star-rated hotels. The element "accommodation" is explained by the number of corporate units in the accommodation industry, and the element "traffic"

is explained by two variables, namely, the volume of transporting passengers and the number of travel agents. The composite variable was refined into the affective variable and cognitive variable [51]. Figure 1 shows the corresponding correlation between tourist destination image dimensions and antecedent variables.

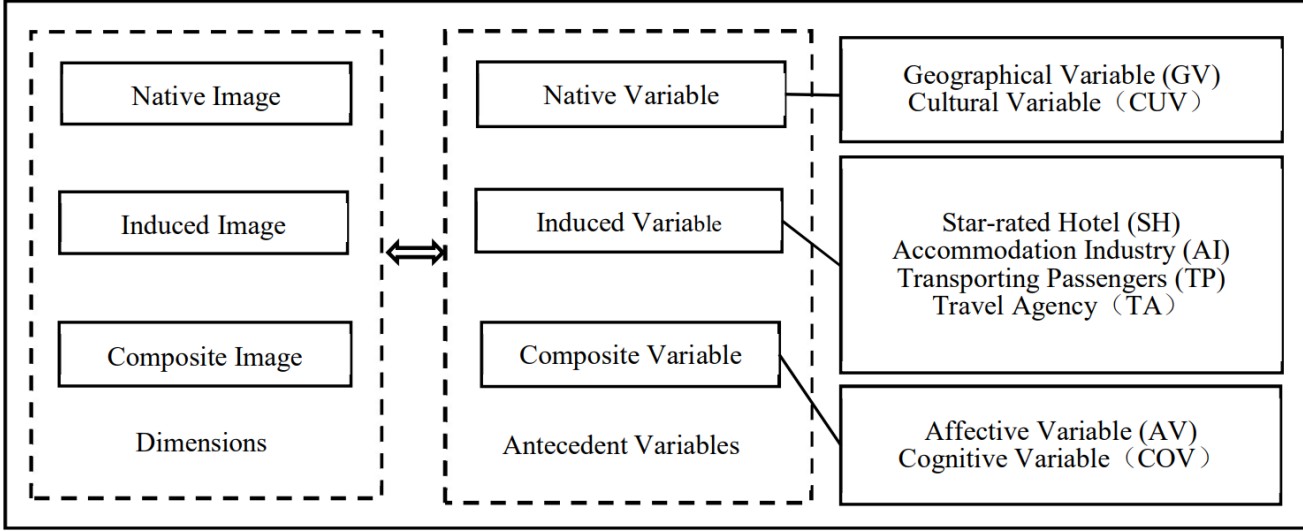

**Figure 1.** Correspondence between dimension and antecedent variable.

*4.3. Data Source*

Notably, the native and the induced variables used in this study were obtained through secondary sources, while the data of the composite variables were obtained through a questionnaire survey conducted from July to August 2020. To facilitate the analysis, a convenience sample of 100 respondents were approached to complete the survey. These respondents were interested and volunteered to participate in this study. To fill out the questionnaire, the respondents must meet two criteria: (i) they must have been to the city in this study, and (ii) they must be adult tourists. We also stipulate that they must fill out the questionnaire completely and truthfully. If they do not meet these requirements (for example, with missing items), they will be reminded to fill them out again until a questionnaire was fully completed. Hence, a full response (100%) was obtained. Due to the COVID-19 pandemic and its associated restrictions, we chose to publish and analyze the questionnaire on the website (https://www.wjx.cn, accessed on 15 August 2020). Taking Qingdao City in Shandong Province as an example, the website of the questionnaire is as follows: https://www.wjx.cn/wjx/design/previewmobile.aspx?activity=86913824&s=1 (accessed on 29 August 2020).

Among the native variables, GV was calculated by weighting the total score based on the grade and quantity of natural world heritage and natural tourism resources above the triple-A class (3A). The CUV was calculated by weighting the total score based on the grade and quantity of humanistic world heritage and humanistic tourism resources above 3A. The data for GV and CUV were obtained from the official website of the Shandong Provincial Department of Culture and Tourism [52] and the World Heritage List in China 2019 [53].

Among the induced variables, SH and TA were obtained from the Culture and Tourism Bureau of each city in Shandong Province [54,55], and AI and TP were obtained from the Shandong Provincial Statistical Yearbook 2020 [56].

Among the composite variables, AV and COV were obtained from a questionnaire for each city and were calculated by the average score of the item. The composite image was the synthesis of tourists' impressions, understandings, and evaluations of tourist destinations before, during, and after traveling [3], emphasizing tourists' subjective feelings. Consumer behavior explores the perception image of tourists from two parts—affection

and cognition [50]. For the questionnaire design, we referred to the related literature to determine the final measurement scale [57]. On the basis of the measurement dimension of destination image proposed by Echtner and Ritchie, the questionnaire design was modified according to the actual situation [58]. The measurement of the cognitive image covered 37 items in 8 aspects: accommodation conditions, information and transportation, business management, communication activities, sightseeing and entertainment activities, scenic environment, shopping activities, and crowding degree. The affective image was measured by four items of "this trip made me very happy", "this trip made me drowsy", "this trip made me very excited", and "this trip made me very annoyed". Table 1 shows the statistical data.

**Table 1.** Statistical data for variables.

| Cities | GV (Score) | CUV (Score) | SH (Score) | AI (Number) | TP (Millions) | TA (Number) | AV (Score) | COV (Score) |
|---|---|---|---|---|---|---|---|---|
| Qingdao | 272 | 491 | 320 | 368 | 47.6 | 505 | 142 | 16 |
| Ji'nan | 143 | 187 | 255 | 314 | 31.9 | 323 | 158 | 20 |
| Zibo | 148.5 | 233 | 80 | 141 | 6.1 | 202 | 138 | 16 |
| Zaozhuang | 127 | 142 | 56 | 101 | 25.7 | 179 | 159 | 18 |
| Yantai | 275.5 | 302.5 | 244 | 273 | 59.1 | 252 | 148 | 20 |
| Weifang | 254.5 | 286 | 161 | 185 | 60.8 | 246 | 161 | 19 |
| Ji'ning | 220 | 499 | 133 | 344 | 40.7 | 243 | 151 | 16 |
| Linyi | 223 | 313 | 87 | 135 | 51.6 | 155 | 157 | 16 |
| Tai'an | 187.5 | 230 | 76 | 123 | 29.8 | 107 | 157 | 20 |
| Liaocheng | 92 | 160.5 | 24 | 80 | 20.1 | 117 | 143 | 16 |
| Heze | 39.5 | 62 | 21 | 201 | 49.7 | 121 | 137 | 16 |
| Dezhou | 55.5 | 230 | 57 | 95 | 19.1 | 101 | 151 | 16 |
| Binzhou | 140 | 296.5 | 21 | 59 | 10.6 | 182 | 152 | 16 |
| Dongying | 72.5 | 275 | 80 | 45 | 6.7 | 290 | 147 | 19 |
| Weihai | 175 | 163.5 | 168 | 152 | 33.8 | 234 | 163 | 20 |
| Rizhao | 188 | 165 | 46 | 50 | 25.3 | 112 | 158 | 17 |
| Laiwu | 54.5 | 62 | 26 | 34 | 1.9 | 33 | 155 | 18 |

*4.4. Analysis Process*

We according to the process of "setting anchors"—"conditional analysis"—"configuration analysis"—"result analysis" using fsQCA (3.0) software [46,47].

### 4.4.1. Setting Anchors

First, the outcome and antecedent variables should be calibrated, that is, anchors should be set for the variables. Based on the research objectives of this study and the related literature, a total of three anchors were set [59].

The first anchor was the cross-point, represented by 0.5, which was the most ambiguous point of membership and non-affiliation, based on the average data of each variable. The second and third anchors were the complete demarcation points, represented by 0.05 and 0.95, which were completely non-affiliated and fully affiliated points, respectively. The values were sorted per the proportion of the corresponding demarcation points in the order of the size of each variable. Table 2 shows the variable calibration anchors.

### 4.4.2. Conditional Analysis

Necessity analysis refers to exploring the extent to which the result set constitutes a subset of the condition set; that is, if a condition always occurs when a specific result exists, then this condition is a necessary condition for the existence of the result [45]. Broadly, the minimum conformance score to identify the necessary conditions is 0.9. The data in Table 3 show that the lack of tourist hotel variables is a necessary condition for a non-high tourist image.

**Table 2.** Anchors for variables.

| Category of Variable | Set's Goal | | Anchors | | |
| --- | --- | --- | --- | --- | --- |
| | | | Non-Affiliated Point | Cross-Point | Fully Affiliated Point |
| Outcome Variables | TTL | High | 62.00 | 71.53 | 83.00 |
| | GV | High | 62.00 | 79.47 | 99.00 |
| | CUV | High | 67.00 | 76.06 | 99.00 |
| | SH | High | 63.00 | 71.24 | 91.00 |
| Antecedent Variables | AI | Much | 61.00 | 74.59 | 97.00 |
| | TP | Big | 62.00 | 79.00 | 98.00 |
| | TA | Much | 65.00 | 73.76 | 84.00 |
| | AV | Strong | 61.00 | 81.94 | 96.00 |
| | COV | Strong | 70.00 | 75.88 | 90.00 |

**Table 3.** Test of a necessary condition.

| Antecedent Variables | | High Image | | Non-High Image | |
| --- | --- | --- | --- | --- | --- |
| | | Consistency | Coverage | Consistency | Coverage |
| Native Variable | GV/~GV | 0.388 | 0.436 | 0.823 | 0.897 |
| | CUV/~CUV | 0.359 | 0.493 | 0.809 | 0.750 |
| | SH/~SH | 0.335 | 0.443 | 0.874 | 0.830 |
| Induced Variable | AI/~AI | 0.401 | 0.457 | 0.778 | 0.837 |
| | TP/~TP | 0.394 | 0.483 | 0.811 | 0.816 |
| | TA/~TA | 0.229 | 0.356 | 0.952 | 0.816 |
| Composite Variable | AV/~AV | 0.588 | 0.582 | 0.546 | 0.684 |
| | COV/~COV | 0.364 | 0.466 | 0.665 | 0.647 |

Note: "~" means "missing condition".

### 4.4.3. Configuration Analysis

In this study, as the case size was small, the frequency was set to 1, and the frequency threshold was set to retain 82% of the total number of cases [45]. In addition, the robustness test was conducted by changing the consistency threshold value (increased from 0.8 to 0.85); the result indicated robustness.

As the QCA method has the characteristics of causal asymmetry [45], that is, whether a specific result appears or not, requires different "cause combinations" to explain separately. In this study, the reason for high tourist image is not the reverse condition that leads to the non-high tourist image. Thus, a configuration analysis must be performed on the high and non-high tourist images. In the high image configuration analysis, we assumed that the appearance of each antecedent variable may enhance the tourist destination image. According to this hypothesis, we further selected to include only prime implicants that appear under the above conditions: SH and GV*CUV. In the non-high image configuration analysis, we assumed that the absence of each antecedent variable may inhibit the promotion of the tourist destination image. Table 4 shows the results of the configuration analysis.

In the analysis of the high tourism image configuration, the consistency index of the overall plan was 0.911, which further showed that the four configurations covering most cases were sufficient conditions to promote the image of tourism purposes. The coverage of the overall plan was 0.763, suggesting that four configurations explained about 76% of the reasons for the high tourist image. In the analysis of non-high tourism image configuration, the consistency index of the overall plan was 0.964, which further showed that the five configurations covering most cases were sufficient conditions to inhibit the promotion of tourism purpose image. Furthermore, the coverage of the overall plan was 0.763, suggesting that five configurations explained about 78% of the reasons for the non-high tourist image.

**Table 4.** Configurational analysis.

| Combination of Variable | | High Image Configuration | | | | Non-high Image Configuration | | | | |
|---|---|---|---|---|---|---|---|---|---|---|
| | | C1 | C2 | C3 | C4 | C5 | C6 | C7 | C8 | C9 |
| Native Variable | GV | ● | ● | ● | | ◎ | | ◎ | ◎ | ◎ |
| | CUV | | ● | ● | | | ◎ | | ◎ | ◎ |
| Induced Variable | SH | | | ● | ● | ◎ | ◎ | ◎ | ◎ | ◎ |
| | AI | | | • | • | ◎ | ◎ | ◎ | | ◎ |
| | TP | | • | • | • | ◎ | ◎ | ◎ | | ◎ |
| | TA | | | • | • | | ◎ | ◎ | ◎ | ◎ |
| Composite Variable | AV | • | • | | • | ◎ | | | ◎ | |
| | COV | ● | | | ● | | ◎ | ◎ | ◎ | |
| Consistency | | 0.853 | 0.976 | 1.000 | 0.991 | 0.960 | 0.982 | 0.982 | 0.982 | 0.965 |
| Raw Coverage | | 0.380 | 0.426 | 0.472 | 0.283 | 0.413 | 0.398 | 0.410 | 0.406 | 0.522 |
| Unique Coverage | | 0.068 | 0.074 | 0.167 | 0.018 | 0.066 | 0.039 | 0.039 | 0.101 | 0.134 |
| Consistency of the overall Plan | | 0.911 | | | | 0.964 | | | | |
| Coverage of the Overall Plan | | 0.763 | | | | 0.780 | | | | |

Note: •, a certain condition appears; ◎, a certain condition does not appear. The big circle indicates the core condition, the small circle indicates the supplementary condition, and the blank indicates the condition that has no influence. C1–C9 represent nine different configurations.

#### 4.4.4. Result Analysis

The results of the configuration analysis in Table 4 showed that the fsQCA method is applicable in the image research of tourist destinations. Promoting or hindering the tourism image is beneficial for this study. The configuration of promotion should be utilized, whereas the configuration of obstacles and its shortcomings should be supplemented. Thus, the nine configurations can become measures to improve tourist destination image. We interpreted the types and paths of tourist destination image in Shandong from the perspectives of "segmentation–integration" and "enhancing strengths–supplementing weaknesses", as shown in Figure 2.

(1)    From the perspective of "segmentation–integration"

First, Shandong Province was divided into 17 cities to analyze tourist destination image. The promotion path of its tourist destination image might also differ due to different cities. According to the output result of fsQCA (3.0) software, the 17 cities were divided into 9 groups, and the cities in the same group had the same path. As the fsQCA method considered all paths as a complete research object, all configurations must be integrated to conduct a comprehensive analysis of the overall image of the tourist destination in Shandong Province. Alternatively, only one measure can enhance the tourism destination image in Shandong Province, that is, specific paths and cities should be analyzed in detail, due to the causal complexity of tourist destination image.

(2)    From the perspective of "enhancing strengths–supplementing weaknesses"

The path to enhance the tourist destination image in Shandong Province can be segmented into two types—"enhancing strengths" and "supplementing weaknesses", which have four and five configurations, respectively. Thus, nine routes were available to improve the tourism image of Shandong Province, with corresponding representative cities.

The configuration of "enhancing strengths" included four paths. C1 indicates that as long as the combination of "GV*AV*COV" appears, it will help promote the tourism image of Tai'an and Weihai Cities. Alternatively, the high tourism image of Tai'an and Weihai Cities benefited from their rich natural tourism resources and the high degree of tourists' cognition, as well as the promotion of tourists' affection. C2 implies that the combination

of "GV*CUV*TP*AV" will help promote the tourism image of Linyi City. Of note, the high tourism image of Linyi City benefited from its rich tourism resources, as well as the joint promotion of transporting passenger and tourists' affection. H3 indicates that the combination of "GV*CUV*SH*AI*TP*TA" will help promote the tourism image of Qingdao, Yantai, and Ji'ning Cities. A further explanation is as follows: The high tourism image of the three cities is driven by their rich tourism resources and the high weight score of star-rated hotels, as well as the joint promotion of the corporate units in the accommodation industry, transporting passenger, and travel agencies. C4 implies that the combination of "SH*AI*TP*TA*AV*COV" will help promote the tourism image of Ji'nan City. In other words, its high tourism image benefited from the high weight score of star hotels and the high degree of tourists' cognition, as well as the joint promotion of some factors that corporate units in the accommodation industry, transporting passengers, travel agencies, and tourists' affection.

The configuration of "supplementing shortcomings" included five paths. C5 implies that as long as the combination of "~GV*~SH*~AI*~TP*~AV" is missing, it might inhibit the promotion of Dongying and Zibo's tourism image. A further explanation is as follows: If Dongying and Zibo Cities do not want a low tourism image, then they must focus on the combined development of five antecedent variables—natural tourism resources, star-rated hotels, corporate units in the accommodation industry, transporting passengers, and tourists' cognition. Particularly, natural tourism resources and star-rated hotels are indispensable. C6 implies that as long as the combination of "~CUV* ~SH*~AI*~TP*~TA* ~COV" is missing, it might inhibit the promotion of tourism image of Rizhao City. A further explanation is as follows: If Rizhao does not want a low tourism image, then it should focus on the combined development of six antecedent variables, namely, cultural tourism resources, star-rated hotels, corporate units in the accommodation industry, transporting passengers, travel agencies, and tourists' cognition. Among them, cultural tourism resources, star-rated hotels, and tourists' cognition are indispensable. C7 implies that as long as the combination of "~GV*~SH*~AI*~TP*~TA*~COV" is missing, it might inhibit the improvement of Binzhou's tourism image. A further explanation is as follows: If Binzhou does not want to have a low tourism image, then it should focus on the combined development of six antecedent variables: natural tourism resources, star-rated hotels, corporate units in the accommodation industry, transporting passengers, travel agencies, and tourists' cognition. Particularly, natural tourism resources, star-rated hotels, and tourists' cognition are indispensable. C8 implies that as long as the combination of "~GV*~CUV*~SH*~TA*~AV* ~COV" is missing, it might inhibit the improvement of Heze's tourism image. A further explanation is as follows: If Heze City does not want a low tourism image, then it should focus on the combined development of tourism resources, star-rated hotels, travel agencies, and tourists' affection and cognition. Particularly, tourism resources, star-rated hotels, and tourists' cognition are indispensable. C9 implies that as long as the combination of "~GV*~CUV*~SH*~AI*~TP*~TA" is missing, it might inhibit the improvement of tourism image of Laiwu and Zaozhuang. A further explanation is as follows: If the two cities do not want a low tourism image, then they should focus on the combined development of tourism resources, star-rated hotels, corporate units in the accommodation industry, transporting passengers, and travel agencies. Among them, tourism resources and star-rated hotels are indispensable.

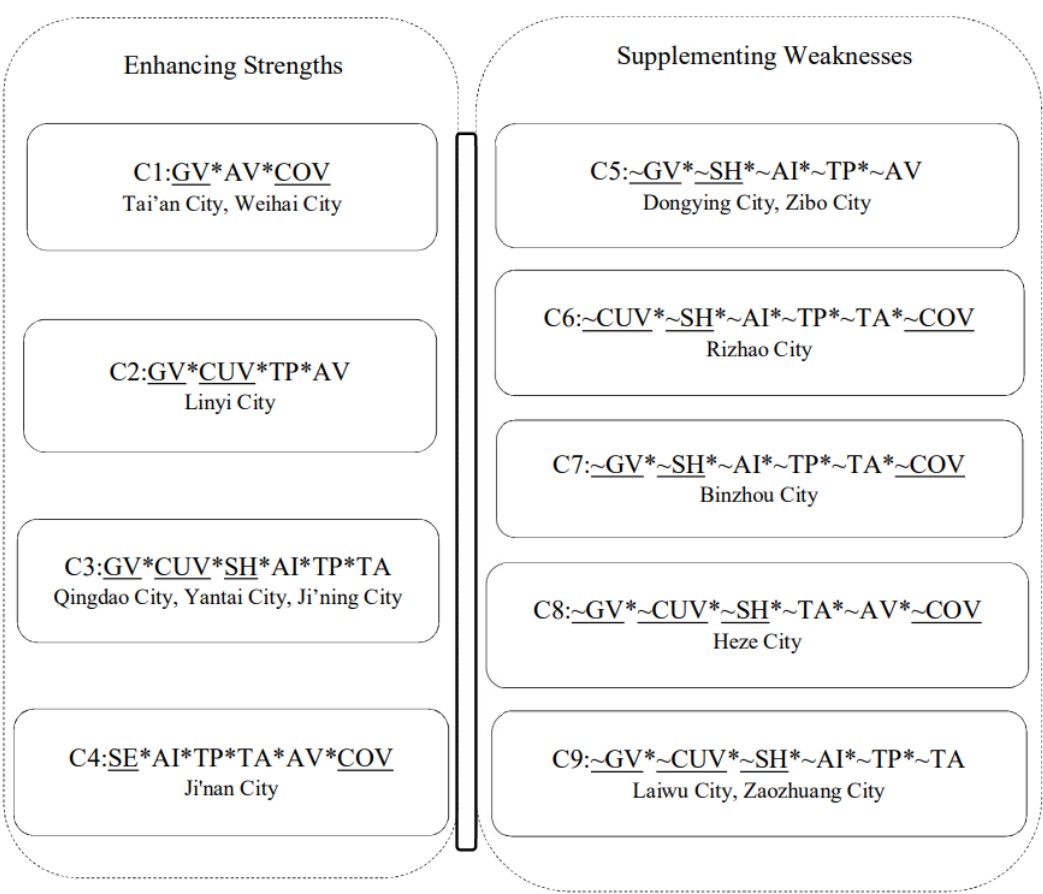

**Figure 2.** Types and paths to enhance tourism image in Shandong Province. Note: with "_", a core condition, without "_", a supplementary condition; "*", indicates the juxtaposition between variables; "~" indicates that the condition is missing.

## 5. Discussion and Limitations

This study establishes that the fsQCA method is applicable and feasible in studying tourist destination images, and the ideal analysis results are obtained. Moreover, it makes up for two limitations in the literature review. Regarding the first limitation, this study combines the dimensions of tourist destination image with the evaluation method from the research content and transforms those dimensions into variables in the evaluation method for research. Regarding the second limitation, the sustainable tourist destination image was examined from the viewpoint of configuration, that is, different combinations of different dimensions (or variables) were considered a complete research object, and the logical thinking of "segmentation–integration" and "strengths–complements" promotes the tourist destination image. Accordingly, the changeable configuration analysis could be realized, and the organic unity of the part and the whole could be attained, which could inspire future image management and planning direction.

The configuration analysis presented in Table 4 reveals that most of the native variables were core conditions, and most of the induced variables were supplementary conditions. This result further reveals that the rich tourism resources in Shandong Province are the key to enhancing the sustainable tourist destination image, and tourism conditions, such as food, housing, and transportation, play a decisive auxiliary role. We need to determine which combination of factors can improve or hinder the tourist destination image in which cities. Accordingly, targeted measures should be taken for each city in the region according to local conditions to achieve accurate governance. This method not only prevents the phenomenon of "one size fits all" from giving consideration to one and losing the other, but it also helps further the scientific understanding and overall explanation of the causal complexity of the regional tourism image and is more conducive to enhancing

the sustainable tourist destination image based on saving the allocation of resources as much as possible. Regarding the limitations of the research, they can be identified based on the used methodology [30].

As this study is the first to apply the fsQCA method to tourist destination image, some points merit discussion, including the selection of the secondary dimensions (variables) of tourist destination image. For example, in the inducement image, the selection basis of this study was "diet, accommodation, and traffic"; moreover, it included other variables such as official organization activities, the number of spontaneous network dissemination, and tourism enterprise advertising because the inducement image cannot be exhausted. Only being able to choose from one's own angles can lead to incomplete image configuration of tourist destinations presented. However, if the secondary variables are exhausted for configuration analysis, it violates feasibility and reality. Therefore, further research needs to focus on how to present the most comprehensive and differentiated configuration of tourist destination image.

## 6. Conclusions

On the basis of the configuration analysis, the sustainable tourist destination image of the entire province is promoted with the logical thinking of "segmentation–integration" and "enhancing strengths–supplementing weakness", which can enlighten the image management and planning direction of tourist destinations in the future.

For the managers, including provincial and municipal tourism managers, provincial managers always maintain the overall planning of resource allocation. Moreover, municipal managers no longer act independently in the development of tourism and always put the overall interests first. For example, provincial managers can cancel the ranking of municipal tourism performance to not cause malicious competition among municipal governments. For the government, a series of policies (such as investment attraction and tax preference) can be issued to attract tourism enterprises to register in a city to make up for its disadvantages in developing tourism. For the tourists, especially tourists from other provinces, as long as they enter the province for tourism activities, they can enjoy more convenient and warm linkage tourism services. For example, Ji'nan City has advantages in PT variable, whereas Laiwu City, which is adjacent, shows disadvantages. Thus, the government's overall allocation of resources will enhance tourism satisfaction. For the analyzed region, only from the standpoint of unified coordination and mutual assistance will it be used for reference by other provinces in the aspect of the sustainable tourist destination image.

The assumption is as follows: We consider the provincial tourism department as the "head office", the municipal tourism department as the "branch", and the variables as the resources that the head office can control. With enterprise resource planning as the core, we establish a computer-controlled tourism linkage system to coordinate and control the allocation of resources through the head office. For example, in configuration C1, Tai'an and Weihai Cities have relatively high tourist destination image compared with the whole Shandong Province, which benefits from the strong support by GV, AV, and COV conditions. When coordinating the resource allocation, the head office should not consider decreasing the support for these conditions. For configuration C5, Dongying City and Zibo City have relatively low tourist destination image compared with the whole Shandong Province because they are not strongly supported by GV, SH, AI, TP, and AV conditions at the same time; thus, the head office should consider increasing support for these conditions when coordinating resource allocation. If the head office has limited resources, it can guide branches to perform mutual resource assistance by issuing policies, holding activities, and other means. Even a crisis warning can be set in this system to wait for the support of the head office and other branches. In this manner, ensuring the tourism development of each subordinate city is equivalent to promoting the sustainable tourism development of the whole province.

**Author Contributions:** Conceptualization, H.X. (Honggen Xiao); methodology, Y.L.; software, Y.L.; validation, N.B.; formal analysis, H.X. (Honggen Xiao); investigation, J.L.; resources, H.X. (Hui Xia); data curation, L.K., H.Y.; writing—original draft preparation, Y.L.; writing—review and editing, Y.L.; visualization, H.Y.; supervision, N.B.; project administration, N.B.; funding acquisition, H.X. (Hui Xia). All authors have read and agreed to the published version of the manuscript.

**Funding:** This research was funded by National Social Science Fund of China, grant number 20BJY131.

**Conflicts of Interest:** The authors declare no conflict of interest.

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
