# Peer review of "Configuration-Based Promotion: A New Approach to Destination Image Sustainability"

_sustainability, doi:10.3390/su132112174_

Round 1

Reviewer 1 Report

Dear Authors,
I have the following remarks about your article.

1. The Introduction section should be reorganized and improved because a scientific article must describe in this section the following issues:
- the research gap: please present the identified gap within the previous research studies and what you want to cover by your own research;
- the research question: present what your manuscript answers to? What question(s) do you intend to clarify in the field of knowledge?
- the article's goal: here you have to describe the main goal of your research;

2. In the section 2. Literature Review, sub-section "2.2. Evaluation method of the tourits destinations' image", rows 110-112 you shortly present some evaluation methods. Here I recommend you to cite https://doi.org/10.3390/jtaer16040056 because this article is about SEM (Structural Equation Modelling) method, https://doi.org/10.1088/1755-1315/511/1/012001 (SEM in tourism), https://www.researchgate.net/publication/273767150_The_Public_Relations_Events_in_Promoting_Brand_Identity_of_the_City (this article is about the process of creating a relevant image).

3. At this moment, the "Literature Review" section seems to be only a descriptive chapter, without any relevant result for the readers.
This chapter seems to be separated from the rest of the article and you don't present a relationship with your research approach.
After the sections 2.1, 2.2 and 2.3, I recommend you to add a new sub-section: 2.4. Research Hypotheses.
Here you have to clearly present and describe your own research questions in a scientific manner. Here you should read and cite http://ecoforumjournal.ro/index.php/eco/article/view/464 because this article offers you a model for organizing the chapters of your manuscript.
The research hypotheses should be validated (or not) by the obtained results.

4. In the section "4.1. Sample Selection" please describe the context of Shandong Province by presenting relevant economic data: number of residents, number of tourist destinations, contribution to GDP etc. This way, the reader will understand the relevance of your research in the general context. 
Between rows 185-188 you write: "To reflect the full “comparison” between samples, sample selection should pursue the principle of maximum heterogeneity, that is, it is best to include both positive and negative samples..."
Please revise this sentence. The English needs revision.

5. Figure 1 needs to be better supported by previous results from literature. At this point, the figure seems more like the authors' own perception, but you need to justify the selected variables and relations by calling previous validated results from literature.
So, I recommend you to improve the section "4.2. Variable Calibration".

6. Between rows 227-232 you write:
"Among the induced variables, the data on the number of star-rated hotels and travel agencies were obtained from the Culture and Tourism Bureau of each city in Shandong, and the data on the number of legal entities and traffic passenger traffic were from the “Shandong Provincial Statistical Yearbook 2020.” The data of composite variables came from a questionnaire for each city and were calculated by the average score of the item."
Please present the numbers: number of hotels, number of travel agencies, populations of the cities etc. Thus, the readers will have a clear image about your data sample.

7. In table 3, you have the header including H1-H9. Please describe what H1, ..., H9 mean.

8. Chapter "5. Conclusions" needs serious improvements. I recommend you to present and describe the following aspects:
- managerial implications (for the managers, for the tourists, for the analyzed region, for the government);
- the future research directions (based on your findings, please develop some realistic research directions). Here you can cite https://doi.org/10.1108/K-03-2021-0197 because this article offers you the opportunity to create a future relationship between tourism and e-tourism.

9. Before chapter 5, I recommend you to insert a new section entitled "Discussions".
Here you should present your results by comparing them to previous existing results in the scientific literature.
This section is very important because the readers can understand your own contributions and their relevance for the field of knowledge.

Kind Regards!

Author Response

Thank you very much.

Reviewer 2 Report

It was with curiosity and interest that I read the article “Promotion path based on configurational perspective: A new approach to the sustainable tourism destination image”. The destination image is an interesting, actual and relevant theme taking in account the worldwide growth of tourist activity and increasingly fierce competition between destinations.

This article complies with almost all the established norms for the articles of this publication in extension and organization, offers interesting results and has a considerable bibliographic reference, interspersing classic references with current works.

However, there are a number of issues that the authors should take into account:

1 – In the Introduction section: there is no identification or deep explanation of the problem or geographic area to be investigated. It is also not properly explained the motivation, purpose of the research and its objectives;

2 – In the Literature Review section: this topic should not be so concentrated in Asiatic references, there are other important studies about “tourism destination image” that must be mentioned to enrich the framing of the theme; point 2.2., line 103, what is IPA analysis? Not all readers are familiar with certain terms, authors should spell out what it is about. The key-word “sustainable tourism destination image” have not been explored in the literature review;

3 – Research process section: this section is somewhat confusing, needs clarification. In 4.1 point, the authors refer to 17 cities but do not mention or locate them on a map, or do not explain why Shandong Province was chosen. Those 17 cities constitute all the cities in this province? If not, what did the authors base themselves on to choose these 17 cities? In other words, why these and not others? This part is not clear to the reader;  4.2 point, line 205, 206 and 212 isn't it "geographical variable and cultural variable" and “affective variable and cognitive variable”?; in figure 1, in the column of the variables (3rd), in the "Induced variable" should not appear the “travel agencies”?, the authors refers in the text four aspects for “induced image” and only three appear in the figure. It would be more understandable if abbreviations also appeared in the text when the variables are referred to; in point 4.3, line 220 and 223, what do the authors refer to when they mention "3A"? The authors also refer to the application of a questionnaire, but it is necessary to understand to whom and when the questionnaires were applied, the authors report that 100 were distributed in each city: to whom? where? It can be insured that there will be 1700 in total, is this sample size representative? What is the sample Universe? The authors should clarify the methodology process to allow replicability and validity; 4.4.3 point, table 3 should be corrected the expressions “High Imge” and “Non-high Imge” to “High Image” and “Non-high Image”;

4 – There's a real discussion section missing that provides a dialogue between previous research and results, the authors just describe the data and results. No exploration is done about the defined key-word “sustainable tourism destination image”. No concrete and effective practical implications and proposals were specified. I suggest that this section be improved.

5 – The authors do not follow the layout rules of the journal in terms of bibliographic references in the text. These should appear in the text with numbers ordered in order of appearance.  I recommend that the authors follow all the instructions for Authors of the journal (https://www.mdpi.com/journal/sustainability/instructions).

6 – Finally, from my point of view English must be revised by a professional.

Author Response

Thank you very much.

Round 2

Reviewer 1 Report

Dear Authors
I appreciate the changes made in the article and its current quality. It is an interesting, topical and useful research.

Congratulations for the scientific content and structure of the article.

Best regards

Author Response

The authors would like to thank Reviewer 1 for your time and effort in helping us improve the quality of this paper.

Thank you very much for your kind comments.

Reviewer 2 Report

The authors greatly improved the text and responded to the indications/suggestions made. I draw attention only to small details regarding formatting that need to be reviewed before publication:

  • line 21: two endpoints after the reference;
  • line 88: two endpoints after the reference;
  • line 90: lacks a space between model(SEM);
  • line 113: a endpoint after the reference and that should have the corresponding number ahead;
  • line 158 and 159: repeated ideas "Hence, we selected all 17 cities in Shandong Province as the research area. We selected 17 cities in Shandong Province as the research area";
  • line 178: lacks a space between website[45]. 
  • Figure 1: the addition of information on the board caused you to stop seeing other information that was hidden;
  • line 222: extra space in " setting anchors";
  • line 223: the endpoint should come after the references"result analysis." [46,47];
  • Table 4: continue to be wrong the expressions “High Iamge” and “Non-high Iamge” that must be corrected to 
    “High Image” and “Non-high Image”;
  • line 335: lacks a space after the endpoint - indispensable.C8 implies;
  • Line 407: extra space before the comma For configuration C5 ,

Author Response

The authors would like to thank Reviewer 2 for your time and effort in helping us improve the quality of this paper.

Thank you very much for your kind comments. 
